# Vitamin A Plasma Levels in COVID-19 Patients: A Prospective Multicenter Study and Hypothesis

**DOI:** 10.3390/nu13072173

**Published:** 2021-06-24

**Authors:** Phil-Robin Tepasse, Richard Vollenberg, Manfred Fobker, Iyad Kabar, Hartmut Schmidt, Jörn Arne Meier, Tobias Nowacki, Anna Hüsing-Kabar

**Affiliations:** 1Department of Medicine B for Gastroenterology, Hepatology, Endocrinology and Clinical Infectiology, University Hospital Muenster, 48149 Muenster, Germany; richard.vollenberg@ukmuenster.de (R.V.); iyad.kabar@ukmuenster.de (I.K.); hepar@ukmuenster.de (H.S.); joernarne.meier@ukmuenster.de (J.A.M.); tobias.nowacki@ukmuenster.de (T.N.); Anna.Huesing-Kabar@ukmuenster.de (A.H.-K.); 2Center for Laboratory Medicine, University Hospital Muenster, 48149 Muenster, Germany; manfred.fobker@ukmuenster.de

**Keywords:** COVID-19, vitamin A, retinol, retinoic acid, ARDS, pneumonia, pandemic, SARS-CoV-2, inflammation

## Abstract

COVID-19 is a pandemic disease that causes severe pulmonary damage and hyperinflammation. Vitamin A is a crucial factor in the development of immune functions and is known to be reduced in cases of acute inflammation. This prospective, multicenter observational cross-sectional study analyzed vitamin A plasma levels in SARS-CoV-2 infected individuals, and 40 hospitalized patients were included. Of these, 22 developed critical disease (Acute Respiratory Distress Syndrome [ARDS]/Extracorporeal membrane oxygenation [ECMO]), 9 developed severe disease (oxygen supplementation), and 9 developed moderate disease (no oxygen supplementation). A total of 47 age-matched convalescent persons that had been earlier infected with SARS-CoV-2 were included as the control group. Vitamin A plasma levels were determined by high-performance liquid chromatography. Reduced vitamin A plasma levels correlated significantly with increased levels of inflammatory markers (CRP, ferritin) and with markers of acute SARS-CoV-2 infection (reduced lymphocyte count, LDH). Vitamin A levels were significantly lower in hospitalized patients than in convalescent persons (*p* < 0.01). Of the hospitalized patients, those who were critically ill showed significantly lower vitamin A levels than those who were moderately ill (*p* < 0.05). Vitamin A plasma levels below 0.2 mg/L were significantly associated with the development of ARDS (OR = 5.54 [1.01–30.26]; *p* = 0.048) and mortality (OR 5.21 [1.06–25.5], *p* = 0.042). Taken together, we conclude that vitamin A plasma levels in COVID-19 patients are reduced during acute inflammation and that severely reduced plasma levels of vitamin A are significantly associated with ARDS and mortality.

## 1. Introduction

Coronavirus disease 2019 (COVID-19) is a novel infectious disease that has been spreading worldwide [1]. The clinical manifestation of COVID-19 can range from asymptomatic infection to critical illness with severe pneumonia, respiratory failure, and death [2]. Worse clinical outcomes are related to dysregulated immune responses in the host, leading to the uncontrolled release of proinflammatory cytokines such as interleukin-6 (IL-6). This cytokine storm mediates the progression of lung damage and respiratory failure in a relevant number of cases [3]. The understanding of host parameters leading to the susceptibility of immune dysregulation is incomplete. A standard therapy has not yet been established. To date, the RNA-polymerase inhibitor remdesivir and the immunosuppressive corticosteroid dexamethasone are the only Food and Drug Administration (FDA) approved drugs for COVID-19 therapy with demonstrated effects on mortality and disease outcome [4,5]. Despite certain therapeutic approaches, the lethality of hospitalized, mechanically ventilated patients remains high [6].

Vitamin A is of special interest in the field of infectious diseases, especially for pulmonary infections. It is crucial for the development of normal lung tissue and tissue repair after injury due to infection [7]; therefore, it may play a role in recovery after severe COVID-19 pneumonia. Vitamin A has immune regulatory functions [8] and positively affects both the innate and adaptive immune cell response [9,10]. Malnutrition leads to relevant incidences of vitamin A deficiency worldwide. Vitamin A deficiency can disrupt vaccine-induced antibody-forming cells and negatively influences immunoglobulin development in the upper and lower respiratory tract [11]. Several studies revealed increased risks of severe illness due to respiratory tract infections in vitamin A-deficient individuals, whereas vitamin A supplementation can reduce the risks of severe illness and death, as was shown for children with measles and in influenza pneumonia in mice models [12,13,14]. The occurrence of severe infections and inflammation can also negatively affect vitamin A status, and several mechanisms such as urinary loss of vitamin A [15], decreased vitamin A hepatic mobilization [16], and reduced intestinal vitamin A absorption [17] during infection have been described.

Data concerning vitamin A plasma levels in COVID-19 patients are lacking. Therefore, this study aimed at characterizing vitamin A plasma levels in acute COVID-19 and analyzed the association of plasma levels with disease severity and outcome. Vitamin A plasma levels were found to be significantly reduced in COVID-19 patients during the acute phase of disease compared to plasma levels in convalescent patients, and a reduction in vitamin A levels correlated significantly with an increase in inflammatory parameters. Critically ill patients showed lower vitamin A levels compared to moderately ill patients and severely reduced vitamin A levels were associated with ARDS and mortality. Further research is necessary to investigate the role of vitamin A as a possible therapeutic agent for COVID-19.

## 2. Materials and Methods

### 2.1. Participant Selection and Patient Samples

This multicenter, prospective observational cross-sectional study included 40 hospitalized patients with laboratory-confirmed SARS-CoV-2 infection (nasopharyngeal swab and test by polymerase-chain reaction), admitted to the University Hospital Muenster and Marien-Hospital Steinfurt in Germany between March and June 2020. Details of medical history and laboratory data were collected. Blood from hospitalized patients was collected during acute phase of disease. Disease severity was defined as critical (presence of acute respiratory distress syndrome [ARDS]; *n* = 22), severe (requiring oxygen supplementation; *n* = 9) or moderate (neither ARDS was present nor oxygen supplementation required; *n* = 9). ARDS was diagnosed according to the Berlin definition (bilateral opacities on chest radiograph, exclusion of other causes of respiratory failure) [18]. COVID-19 patients were categorized according to their condition at the time of blood collection. One blood sample per patient was taken within the first 24 h after admission.

Additionally, 91 individuals with laboratory-confirmed SARS-CoV-2 infection who had recovered from infection were contacted for donation of convalescent plasma in our outpatient clinic. The blood sample was taken after recovery from disease. Of these, 47 were manually selected to match the age and gender of inpatients (Appendix A). This group of patients had only moderate symptoms, and hospitalization was not necessary (Appendix A).

Plasma samples were obtained from each participant after they provided informed consent. The Ethics Committee of Muenster University approved the current study (local ethics committee approval AZ 2020-220-f-S and AZ 2020-210-f-S), and the procedures were in accordance with the Helsinki Declaration of 1975 as revised in 1983. None of the patients received antiviral, experimental, or immunosuppressive therapies. Plasma samples were protected from light and frozen (−80 °C) until measurement.

### 2.2. Vitamin A Measurement

Vitamin A (both free/unbound vitamin A and vitamin A bound to retinol binding protein [RBD]) was assayed on EDTA-plasma using a commercially available high-performance liquid chromatography kit (Chromsystems, Munich, Germany) following the manufacturer’s instructions. Vitamin A levels were given in mg/L. For further analysis, participants were divided in to two groups (Group 1: vitamin A plasma levels ≥ 0.2 mg/L; Group 2: vitamin A plasma levels < 0.2 mg/L) following international guidelines on vitamin A deficiency from the World Health Organization (WHO, clinically relevant nutritional Vitamin A deficiency defined as Vitamin A plasma levels < 0.2 mg/L) [19].

### 2.3. Laboratory Measurements and Validation of the Clinical Status

Clinical laboratory assessment included complete blood count and levels of D-dimer, creatinine, C-reactive protein (CRP), albumin, lactate dehydrogenase (LDH), pseudocholinesterase (PCHe), and alanine aminotransferase (ALT). SAPS II (Simplified Acute Physiology Score [20]) was determined on the day of laboratory measurement and used to characterize the physiological conditions of hospitalized patients.

### 2.4. Data Analysis/Statistics

For continuous variables, we report the median with the interquartile range, and values were compared using the Mann–Whitney U (Wilcoxon) test. For categorical variables, we report absolute numbers and percentages, and values were compared with Chi-square tests of association or Fisher’s exact tests. A Kruskal–Wallis test was conducted to compare more than two groups. To compare subgroups, the Bonferroni correction post hoc test was performed when variance was equal (Levene’s test), and the Games-Howell test was performed when variance was different. The Pearson correlation coefficient was determined to analyze the correlation of vitamin A levels with clinical laboratory parameters. Univariable logistic regression analysis was conducted to determine the association of severely reduced vitamin A plasma levels with the risk of acute respiratory distress syndrome (ARDS) and mortality due to COVID-19 in hospitalized patients (*n* = 40). All the tests were two-tailed, and *p* < 0.05 was considered to indicate a statistically significant difference. All the statistical analyses were performed using SPSS (Version 26IBM Corp., Armonk, NY, USA).

## 3. Results

### 3.1. Cohort Characteristics

Most participants were male (77.8–100%) in all groups. The median interval between symptom onset and sample collection in hospitalized patients was 12 days (IQR 8, 3–17), in outpatients 52 days (IQR 40–75). The median age was 50–58 years and did not significantly differ between groups, nor too did the interval from the first symptom to the collection of the blood sample. Preexisting diseases were prevalent almost exclusively in the group of critically ill patients. SAPS II differed significantly between hospitalized patients, as did inflammatory and blood count parameters (for example C-reactive protein and lymphocyte count) (Table 1). The correlations of the subgroups are shown in Appendix A.

### 3.2. Correlation of Vitamin A Plasma Levels with Laboratory Parameters

In the overall study group (*n* = 87), reduced vitamin A plasma levels correlated significantly with increased levels of the inflammation markers C-reactive protein (*r* = −0.54, *p* < 0.001) and ferritin (*r* = −0.45, *p* < 0.001). Reduced absolute (*r* = 0.4, *p* < 0.001) and relative lymphocyte counts (*r* = 0.43, *p* < 0.001), reduced albumin levels (*r* = 0.65, *p* < 0.001), and elevated lactate dehydrogenase levels (*r* = −0.53, *p* < 0.001) correlated significantly with reduced vitamin A plasma levels. Elevated levels of liver markers (AST: *r* = −0.22, *p* < 0.05; gamma-GT: *r* = −0.29, *p* < 0.01) also correlated significantly with reduced vitamin A levels. Reduced vitamin A levels were associated with reduced pseudocholinesterase (PCHe) levels as a parameter of the liver synthetic capacity (*r* = 0.53, *p* < 0.001) (Figure 1).

### 3.3. Vitamin A Plasma Levels in COVID-19 Patients

Vitamin A plasma levels differed significantly depending on disease severity. Hospitalized patients of all groups (mild, moderate, severe, and critical disease) revealed significantly reduced vitamin A plasma levels compared to convalescent outpatients (*p* < 0.01 to *p* < 0.001). In hospitalized patients, the vitamin A plasma levels of critically ill patients were significantly reduced compared to patients with moderate disease (*p* < 0.05). Patients with severe disease also had reduced plasma levels compared to patients with moderate disease, but this finding did not reach statistical significance (median: 0.32 vs. 0.48 mg/L) (Figure 2).

### 3.4. Association of Severely Reduced Vitamin A Plasma Levels with the Development of ARDS and Mortality

As described in the methods section, Vitamin A plasma levels <0.2 mg/L were considered clinically relevant reduced. In the overall study group, 14% (*n* = 11/87) of participants revealed clinically relevant reduced vitamin A plasma levels. Looking at different patient groups, 41% (*n* = 9/22) of critically ill, 11% (*n* = 1/9) of moderately ill, and 11% (*n* = 1/9) of severely ill patients had vitamin A plasma levels <2 mg/L, while none of the participants in the convalescent group had clinically relevant reduced vitamin A levels (0/47) (Figure 3).

After dividing the hospitalized patients into two groups (Group 1: Vitamin A < 2 mg/L; Group 2: Vitamin A ≥ 2 mg/L), Group 1 patients (*n* = 11) revealed significantly higher C-reactive protein levels (median 13.9 vs. 6 mg/dL, *p* = 0.03), lower absolute lymphocyte counts (median 0.69 Tsd/µL vs. 1.13 Tsd/µL, *p* = 0.01), and lower relative lymphocyte counts (median 9.4% vs. 15.2%, *p* = 0.049) compared to Group 2 (*n* = 29). Patients in Group 1 developed ARDS much more frequently (*p* = 0.038), and mortality was significantly higher (*p* = 0.047) (Table 2). Logistic regression analysis revealed significant associations of clinically relevant reduced vitamin A plasma levels with the development of ARDS (OR = 5.54 [1.01–30.26] *p* = 0.048) and with mortality (OR 5.21 [1.06–25.5], *p* = 0.042) in the hospitalized patient group (*n* = 40).

## 4. Discussion

This study analyzed associations of vitamin A plasma levels with inflammatory parameters, disease severity, and outcomes in moderately, severely, and critically ill COVID-19 patients during the acute phase of the disease and in comparison to convalescent patients.

Age, gender, and time from disease onset to collection of blood sample adequately matched and did not differ significantly between hospitalized groups in the acute phase of the disease. Our cohort characteristics showed significantly higher levels of inflammatory markers (CRP, IL-6, ferritin) and lower lymphocyte counts in patients with critical disease cases compared to moderately and severely ill and convalescent patients. These data are in line with other studies as both increased inflammatory parameters and decreased lymphocyte counts are well-defined markers of active disease and predictive of a severe course of COVID-19 [21]. D-Dimer is also an established predictive marker [22] and its levels were found to be significantly increased along with disease severity in our study cohort.

This study revealed significantly decreased vitamin A plasma levels in the acute phase of moderately, severely, and critically ill patients compared to convalescent patients after recovery from acute disease. In the acute phase, critically ill patients had significantly lower vitamin A plasma levels compared to moderately ill patients. In the overall study cohort, correlation analysis provided evidence that higher inflammatory parameters such as C-reactive protein, ferritin, and albumin significantly correlated with reduced vitamin A plasma levels. This finding supports existing data showing reduced vitamin A plasma levels due to several mechanisms such as urinary loss [15], decreased hepatic mobilization [16], and reduced absorption [17] during infection and acute inflammatory conditions. Moreover, lymphopenia, an established disease activity marker and predictor of worse outcomes in COVID-19 patients [21], significantly correlated with reduced vitamin A plasma levels. Reduced serum albumin levels [23] and elevated LDH levels [24], also established predictors of severe disease and worse outcome, were both significantly correlated with lower vitamin A plasma levels. Liver damage is often found in severely ill COVID-19 patients [25]. Elevated levels of liver enzymes also correlated significantly with reduced vitamin A levels. Interestingly, a reduction in the levels of pseudocholinesterase (PChE), a parameter for liver synthetic capacity, correlated with reduced vitamin A levels. The association with decreased hepatic vitamin A mobilization during acute inflammation remains speculative but conceivable and needs further analysis.

Clinically relevant decreased vitamin A plasma levels (defined as <2 mg/L following the WHO definition for vitamin A deficiency [19]) were almost exclusively found in the acute phase of disease in critically ill patients. To analyze the correlation of severely reduced vitamin A plasma levels with disease outcome, we divided the hospitalized patient cohort in two groups (patients with vitamin A levels ≥2 and <2 mg/L). This subgroup analysis showed that patients with vitamin A plasma levels <2 mg/L developed ARDS at a significantly higher rate (*p* < 0.05), and mortality was also much higher in this group (*p* < 0.05). Levels of C-reactive protein were higher and lymphopenia more distinctive, indicating a more severe disease course [21]. These results provide evidence that the reduction of vitamin A plasma levels in COVID-19 is dependent on disease severity and that plasma levels are reduced due to acute infection and inflammation rather than preexisting vitamin A deficiency and malnutrition, as convalescent patients had normal levels of vitamin A in plasma. Furthermore, malnutrition and vitamin A deficiency are rare in Germany. Nonetheless, vitamin A deficiency due to malnutrition is still one of the most prevalent micronutrient deficiencies worldwide, affecting approximately one-third of preschool-age children, especially in underdeveloped countries. Vitamin A deficiency is associated with increased mortality in children and pregnant women due to severe gastrointestinal and respiratory infections [19,26]. Several studies have highlighted the importance of vitamin A for lung function and development [27]. Other studies confirmed that vitamin A has immune-modulating properties and plays an important role in the immune response to infections, mainly through retinoic acid, its main metabolite. Early randomized clinical trials studying the effect of vitamin A supplementation showed it resulted in a significant reduction of mortality and less severe manifestations of several infectious diseases in cases of vitamin A deficiency due to malnutrition [12,28]. On the contrary, the consequences of reduced vitamin A levels that are due to acute inflammation are not well understood.

Vitamin A influences cellular immunity in a wide variety. A number of studies have shown that vitamin A has a central function in the development and differentiation of dendritic cells (DCs), the most important antigen-presenting cells for activating naive T-cells. DCs express three isotypes of RA receptor and, therefore, directly respond to vitamin A [29]. After receptor activation, DCs drive T-cell differentiation into either anti-inflammatory regulating T-cells (Treg) or proinflammatory effector T-cells, and through this, they maintain homeostasis between anti-inflammatory and proinflammatory stimuli [30]. To resolve infection, vitamin A leads to the migration of effector T-cells to the inflammatory site via the induction of leukocyte-homing receptors such as CCR9 and α4β7 integrin [31]. Vitamin A initiates the production of proinflammatory cytokines such as interferon-gamma to resolve viral infection [32]. Vitamin A also drives the humoral immune response, as it crucially promotes B-cell maturation and antibody responses in viral clearance [33,34]. To limit proinflammatory stimuli, vitamin A promotes the differentiation and extravasation of anti-inflammatory Treg cells to the site of inflammation [35]. Taken together, vitamin A can promote both proinflammatory and anti-inflammatory cellular immune responses.

In COVID-19, immune imbalance and the disruption of T-cell responses may be important drivers of severe disease. Real-world data suggest a strong antiviral T-cell response after infection with SARS-CoV-2. The majority of SARS-CoV-2-specific T-cells show an effector phenotype with the dominant production of antiviral proteins, such as interferon-gamma, to terminate infection [36]. However, in patients with severe and critical disease, T-cells exhibit lower levels of antiviral interferon-gamma compared to patients with mild disease [37]. This is known as COVID-19-associated “T-cell exhaustion” and may lead to impaired viral clearance. The resulting massive replication of SARS-CoV-2 and viral infection of cells and organs and subsequent viral release from dying cells is considered as an initial driver of cytokine storming, leading to uncontrolled immune reactions and often fatal outcomes due to multiple organ failure [38]. Furthermore, levels of anti-inflammatory Treg cells in critically ill COVID-19 patients are lower than those in patients with mild disease [39], possibly leading to further uncontrolled immune reactions. As described above, vitamin A influences the production of interferon-gamma through effector T-cells as well as the differentiation of anti-inflammatory Treg cells. Our study results show a significant reduction of vitamin A levels in critically ill COVID-19 patients. Owing to these study results, vitamin A should be considered a relevant agent in maintaining immune balance in mild and moderate COVID-19. Immune imbalance and disruption of antiviral T-cell responses in cases of severe and critical COVID-19 may, in part, be driven by decreased vitamin A plasma levels during the acute phase.

This study has some limitations. First, this study can only reveal descriptive data and does not prove causality. Further studies are needed to clarify whether COVID-19 disease course worsens as a result of Vitamin A reduction in patients serum, or Vitamin A levels decrease as a consequence of severe COVID-19 and whether this reduction itself has an impact on disease course. Second, it has to be pointed out that in this study both unbound vitamin A and vitamin A bound to retinol binding protein (RBP) were measured. It is known that hepatic synthesis of RBP is reduced following acute phase reactions in terms of prioritization of the liver for synthesis of acute phase proteins such as CRP and ferritin [40]. Further studies including analysis of free (unbound) vitamin A in plasma are needed to clarify whether vitamin A reduction in plasma in COVID-19 patients is a consequence of reduced RBD synthesis or free (unbound) vitamin A is reduced as well. Third, the sample size in cohort subgroups is small, which can lead to bias especially in analyses for associations between Vitamin A levels and ARDS/mortality. The small number of events in univariate analyses could possibly lead to confounding factors and lead to misinterpretation. Taking these limitations into account the consequences of reduced vitamin A plasma levels in COVID-19 patients require further research to investigate possible therapeutic approaches of vitamin A supplementation during acute infection. Finally, controlled prospective studies are needed to investigate the therapeutic effect of vitamin A supplementation in cases of acute COVID-19.

## Figures and Tables

**Figure 1 nutrients-13-02173-f001:**
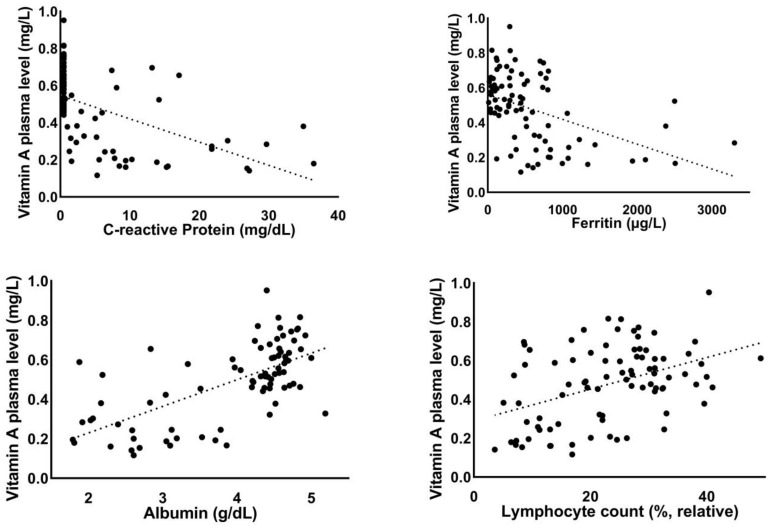
Correlation of vitamin A plasma levels with laboratory parameters.

**Figure 2 nutrients-13-02173-f002:**
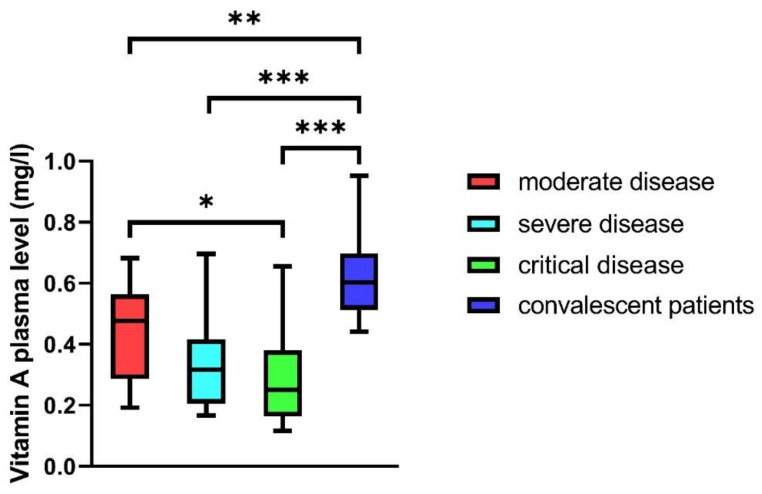
Vitamin A plasma levels in patients with moderate, severe, and critical disease, and in convalescent patients; the Wilcoxon rank test showed significant differences (*p* < 0.001). Subgroups were tested for differences in the Bonferroni correction (* *p* < 0.05, ** *p* < 0.01, *** *p* < 0.001). Mann–Whitney U test was used to compare groups (* *p* < 0.05, ** *p* < 0.01, *** *p* < 0.001).

**Figure 3 nutrients-13-02173-f003:**
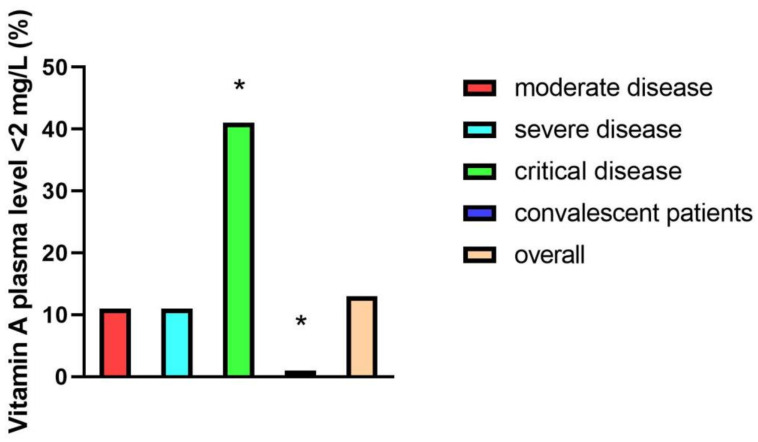
Percentages of patients with vitamin A plasma levels <2 mg/L in different patient groups: moderate disease (*n* = 9), severe disease (*n* = 9), critical disease (*n* = 22), and convalescent patients (*n* = 47). Representation of significances according to Bonferroni correction (* *p* < 0.05).

**Table 1 nutrients-13-02173-t001:** Cohort characteristics; differences were calculated via the Kruskal–Wallis test; BMI = body mass index, SAPS II = Simplified Acute Physiology Score II; IQR = interquartile range; LDH = lactate dehydrogenase; PCHe = pseudocholinesterase, ALT = alanine aminotransferase; abs. = absolute; n.d. = not defined.

	Moderate Disease (*n* = 9)	Severe Disease (*n* = 9)	Critical Disease (*n* = 22)	Convalescent Patients (*n* = 47)	*p*-Value
Age, median (min–max)	54 (30–81)	50 (39–73)	58 (41–82)	54 (41–70)	0.24
Gender, male (%)	77.8	100.0	90.9	97.9	0.29
BMI, median (IQR)	24 (23–26)	24 (23–26)	27 (24–30)	26 (24–28)	0.05
Interval from first symptom to acquisition of blood sample in days, median (IQR)				52 (40–75)	
	8 (6–14)	13 (8.5–17)	12 (10–22)		0.15
Cardiovascular disease (abs.)	1	0	4	0	
Respiratory disease abs.)	0	0	2	0	
Kidney insufficiency (abs.)	0	0	0	0	
Metastatic neoplasm (abs.)	0	0	0	0	
Diabetes (abs.)	0	0	1	0	
Hematologic malignancy (abs.)	2	0	4	0	
Death (abs.)	0	0	9	0	
SAPS II, median (IQR)	15 (13–25)	19 (13–22)	54 (35–72)	n.d.	<0.001
Leukocytes × 10^9^/L, median (IQR)	4.4 (3.4–6.4)	5.4 (3.8–7.6)	9.2 (5.8–11)	5.4 (4.9–6.8)	0.21
Lymphocytes (rel., %), median (IQR)	19.5 (12.6–29.9)	21.3 (15.0–22.8)	10.3 (7.3–14.0)	29.0 (25.3–32.5)	0.003
D-Dimer (mg/L), median (IQR)	0.75 (0.32–2.31)	0.76 (0.55–2.02)	2.56 (1.42–7.42)	n.d.	0.005
Creatinine (mg/dL), median (IQR)	1.0 (0.9–1.2)	0.8 (0.8–1.0)	1 (0.6–1.7)	1 (0.9–1)	0.36
Ferritin (µg/L), median (IQR)	449 (200–665)	692 (370–938)	917 (665–1560)	188 (89–325)	0.003
Interleukin-6 (pg/mL), median (IQR)	16 (10–30)	30 (17–70)	107 (39–239)	2 (2–2)	<0.001
Procalcitonin (ng/mL), median (IQR)	0.11 (0.07–0.18)	0.08 (0.07–0.12)	0.64 (0.18–2.04)	0.05 (0.04–0.07)	<0.001
C-reactive protein (mg/dL), median (IQR)	1.6 (0.5–3.2)	6 (3.3–9.4)	14.8 (6.2–24.8)	0.5 (0.5–0.5)	<0.001
PCHe (U/L), median (IQR)	7746 (5524–9193)	6960 (6173–8321)	3668 (2749–4788)	8699 (7754–10082)	<0.001
Gamma-GT (U/L), median (IQR)	43 (30–126)	40 (30–60)	113 (54–185)	29 (21–47)	<0.001
ALT (U/L), median (IQR)	26 (22–46)	33 (29–58)	41 (29–67)	29 (24–40)	0.079
Albumin (g/dL), median (IQR)	3.9 (3.3–4.5)	3.8 (3.5–4.4)	2.3 (3.0–2.7)	4.6 (4.4–4.7)	<0.001
Vitamin A (mg/L), median (IQR)	0.48 (0.29–0.56)	0.32 (0.21–0.42)	0.25 (0.16–0.38)	0.60 (0.51–0.69)	<0.001

**Table 2 nutrients-13-02173-t002:** Clinical outcomes and inflammation parameters based on vitamin A plasma levels; differences were calculated using the Mann–Whitney U test or Fisher’s exact test (both 2-tailed); SAPS II = Simplified Acute Physiology Score II; BMI = body mass index; ARDS = acute respiratory distress syndrome; IQR = interquartile range; LDH = lactate dehydrogenase; PCHe = pseudocholinesterase, ALT = alanine aminotransferase.

	Vitamin A < 2 mg/L (n = 11)	Vitamin A ≥ 2 mg/L (n = 29)	*p*-Value
Age, median (min–max)	52.6 (30–66)	57.1 (33–82)	0.52
Gender, male (%)	91	90	0.56
BMI, median (IQR)	25 (24–28)	25 (23–28)	0.51
Interval from first symptom to acquisition of blood sample in days, median (IQR)	11 (9–12)	13 (8–17.5)	0.42
Preexisting disease (%)	45	48	0.45
ARDS (abs.)	9	13	0.038
Death (abs.)	5	4	0.047
SAPS II score, median (IQR)	43 (22–61)	28 (15.5–55.5)	0.39
Leukocytes × 10^9^/L, median (IQR)	5.99 (2.67–11.9)	6.97 (4.56–9.89)	0.56
Lymphocytes (rel., %), median (IQR)	9.4 (7.2–16.9)	15.2 (9.35–22.1)	0.049
Lymphocytes (abs., Tsd/µL), median (IQR)	0.69 (0.47–1.01)	1.13 (0.79–1.38)	0.01
D-Dimer (mg/L), median (IQR)	2.01 (0.78–4.25)	1.78 (0.65–3.57)	0.84
Creatinine mg/dL, median (IQR)			0.15
Ferritin (µg/L), median (IQR)	917 (554–1788)	738 (465–1008)	0.39
Interleukin-6 (pg/mL), median (IQR)	88 (37–199)	33 (16–95)	0.05
Procalcitonin (ng/mL), median (IQR)	0.31 (0.12–0.80)	0.13 (0.08–0.7)	0.3
C-reactive protein (mg/dL), median (IQR)	13.9 (8.5–26.9)	6 (1.95–13.7)	0.03
PCHe (U/L), median (IQR)	4082 (3012–5643)	5555 (3566–7814)	0.15
Gamma-GT (U/L), median (IQR)	116 (40–211)	55 (34–130)	0.13
ALT (U/L), median (IQR)	44 (39–70)	31 (26–48)	0.11
Albumin g/dL, median (IQR)	2.7 (2.2–3.3)	3.1 (2.2–4.0)	0.25
Vitamin A (mg/L), median (IQR)	0.17 (0.16–0.19)	0.38 (0.28–0.52)	<0.0001

## Data Availability

Data cannot be made public as personal patient data are included.

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
