# Peer review of "Vitamin A Plasma Levels in COVID-19 Patients: A Prospective Multicenter Study and Hypothesis"

_nutrients, 2021, doi:10.3390/nu13072173_

Round 1

Reviewer 1 Report

General comment on the whole manuscript: It is a great article in the research of COVID-19. Nowadays, research about natural compounds, included vitamins, have a great interest to clarify the development of this illness and its use could improve the patients’s health and quality of life.

Introduction section: It is adequate and references are actualized.

Materials and Methods: This multicenter, prospective observational study is clearly described being explained the inform consent and information about Ethics Committee. The extraction and analysis, joint with statistical analysis of vitamin A is OK. The Figures are visually good.

Results sections: Information of results is OK but I think that the type of letter in the Table 1 is different versus type of letter in the manuscript. Can you see this option?

Discussion section: In my viewpoint, the discussion is greatly adequate and it help to clarify the obtained results.

Affiliation section:  I recommend that authors, in affiliation section, add author initials, due to that if there are multiple corresponding authors is better.

Author Contributions: you must add author initials in the author contributions section

Reviewer 2 Report

Summary

In this observational study including 87 SARS-CoV-2 infected individuals from German, the author tried to identify the role of blood vitamin A during the COVID-19 disease progression, including the changes in plasma vitamin A level, and the association between the changes and ARDS and mortality, etc..  

Comments:

  1. Study design: The current study is not a cohort study. Please clarify the study design in the methods.
  2. Diagnosis criteria: What guidelines on the diagnosis of COVID-19 did the authors used for the mild, moderate, severe, or critical COVID-19. It is unclear how the outpatients were diagnosed. Did they have any symptoms or only have mild symptoms based on the criteria you used?  Please make it clear for the readers.
  3. Sample size: The number of patients enrolled in this study is small, especially for some groups.  In the association between the vitamin A level and ARDS and mortality analysis, only 40 inpatients were included, etc.  A small sample size affects the reliability of a study's results which may lead to bias.  
  4. There are some concerns regarding statistical assessments by authors. E.g., the monovarietal analysis causes many confounding factors that lead a misinterpretation. However, they would not be able to perform the multivariate analysis due to the limited number of events. These are critical issues in the study.
  5. Figure 2, for the multiple comparison, how did the authors correct them.  
  6. Table 2, why "2 mg/L" was a cut point for vitamin A level? Please describe in the methods.
  7. Did the authors collect the blood samples for those inpatients after recovery from the disease?  

Reviewer 3 Report

An interesting paper; however, the authors presented their data and analysis in the manner that vitamin A deficiency, as evidenced by low serum vitamin A concentrations, worsens outcomes for patients with covid-19 disease. Unfortunately, this analysis does not confirm these assumptions. The authors attempted to briefly address this issue in the limitations of the study but is woefully inadequate.  I think the authors need to be clearer that this paper is a presentation of a hypothesis and not proving. My suggestions are as follows:

I would suggest the authors consider a title change to read something like: Vitamin A plasma concentrations in COVID-19 patients: A prospective multicenter study and hypothesis.

It should be discussed that during inflammation or critical illness there is a reprioritization of hepatic protein synthesis. Transport protein synthesis is decreased whereas acute phase proteins are increased. C reactive protein concentrations were elevated, particularly for those with severe illness or critical illness. This may explain the decreased vitamin A concentrations as retinol binding protein concentration decreases during acute illness.

Without free (unbound) vitamin A concentrations (which should be included as a limitation of this study), we do not know if a vitamin A deficiency truly exists in the face of acute illness due to a decrease in RBP.

Table 1. Post-hoc pairwise comparisons should be done to ascertain which groups differ from each other for the variables.

Figure 1. Are all these linear regressions necessary (e.g., LDH, ALT, GGT)? CRP, Ferritin, and Albumin are the most interesting and appears to correspond with known reprioritization process.

Figure 2. Using the mann-whitney u test is incorrect for pairwise comparisons (due to an alpha error) unless a Bonferroni correction is applied with non-parametric multiple groups. Dunn’s multiple comparison test via Kruskall Wallis ANOVA can also be used.

Figure 3. Statistical inference should be applied.

Lines 241-245. The relationship that patients with vitamin A less < 2 mg/L developed ARDS at a higher rate is overreaching. One can also suggest that those with ARDS (and more inflammation) had a lower vitamin A concentration. This needs to be clarified.

Round 2

Reviewer 2 Report

The authors have addressed my concerns.